# A Rare Case of Hepatocellular Carcinoma Recurrence in Ovarian Site after 12 Years Mimicking a Hepatoid Adenocarcinoma: Case Report

**DOI:** 10.3390/jcm12072468

**Published:** 2023-03-24

**Authors:** Stefano Restaino, Giulia Pellecchia, Alice Poli, Martina Arcieri, Claudia Andreetta, Laura Mariuzzi, Maria Orsaria, Anna Biasioli, Monica Della Martina, Sergio Giuseppe Intini, Giovanni Scambia, Lorenza Driul, Giuseppe Vizzielli

**Affiliations:** 1Department of Maternal and Child Health, University-Hospital of Udine, P.le S. Maria della Misericordia n°15, 33100 Udine, Italy; restaino.stefano@gmail.com (S.R.); lorenza.driul@uniud.it (L.D.);; 2Gynecology and Obstetrics Clinic, Department of Medicine, University of Udine, Via Colugna n° 50, 33100 Udine, Italy; giulia.pellecchia95@gmail.com (G.P.); alicepoli94@gmail.com (A.P.); 3Department of Biomedical, Dental, Morphological and Functional Imaging Science, University of Messina, 98122 Messina, Italy; 4Department of Medical Oncology, Azienda Sanitaria Universitaria Friuli Centrale (ASUFC)—Ospedale S. Maria della Misericordia, 33100 Udine, Italy; 5Medical Area Department, Institute of Pathological Anatomy, University of Udine, Azienda Sanitaria Universitaria Friuli Centrale, 33100 Udine, Italy; 6Department of General Surgery, Academic Hospital of Udine, University of Udine, 33100 Udine, Italy; 7Institute of Gynaecology and Obstetrics Clinic, Fondazione Policlinico Universitario A. Gemelli IRCCS, 00168 Rome, Italy; 8Department of Medical Area (DAME), University of Udine, Azienda Ospedaliera Universitaria Friuli Centrale, ASUFC, 33100 Udine, Italy

**Keywords:** liver neoplasms, adnexal diseases, genital neoplasms female, diagnosis differential, case reports

## Abstract

Hepatoid carcinoma of the ovary (HCO) is a tumor that resembles, both histologically and cytologically, hepatocarcinoma (HCC) in a patient with a non-cirrhotic liver not involved by the disease. Hepatoid carcinoma is an extremely rare histologic subtype of ovarian cancer and should be distinguished from metastatic HCC. Here, we report the rare case of a 67-year-old woman with ovarian recurrence of HCC 12 years after first diagnosis. The patient was being followed by oncologists because she had been diagnosed with HCV-related HCC (Edmonson and Stainer grade 2, pT2 N0 M0, G2, V1) in 2009. She had undergone surgery for enlarged left hepatectomy to the 4th hepatic segment with cholecystectomy and subsequent placement of a Kehr drain. The preoperative alpha-fetoprotein (AFP) level was 8600 ng/mL, while the postoperative value was only 2.7 ng/mL. At the first diagnosis, no other localizations of the disease, including the genital tract, were found. At the time of recurrence, however, the patient was completely asymptomatic: her liver function was within normal limits with negative blood indices, except for an increased blood dosage of AFP (467 ng/mL), and CA125, which became borderline (37.4 IU/mL). The oncologist placed an indication for a thoracic abdominal CT scan, which showed that the residual liver was free of disease, and the presence of a formation with a solid–cystic appearance and some calcifications at the left adnexal site. The radiological findings were confirmed on level II gynecological ultrasound. The patient then underwent a radical surgery of hysterectomy, bilateral oophorectomy, pelvic peritonectomy, and omentectomy by a laparotomic approach, with the sending of intraoperative extemporaneous histological examination on the annexus site of the tumor mass, obtaining RT = 0. Currently, the patient continues her gyneco-oncology follow-up simultaneously clinically, in laboratory, and instrumentally every 4 months. Our study currently represents the longest elapsed time interval between first diagnosis and disease recurrence, as evidenced by current data in the literature. This was a rather unique and difficult clinical case because of the rarity of the disease, the lack of scientific evidence, and the difficulty in differentiating the primary hepatoid phenotype of the ovary from an ovarian metastasis of HCC. Several multidisciplinary meetings for proper interpretation of clinical and anamnestic data, with the aid of immunohistochemistry (IHC) on histological slides were essential for case management.

## 1. Introduction

Hepatoid adenocarcinoma is a tumor that resembles both histologically and cytologically a HCC in a patient with a non-cirrhotic liver not involved by the disease [1]. It is, by definition, an extrahepatic adenocarcinoma with hepatocyte differentiation [2]. It could arise in different organs such as lung, bladder, kidney, uterus, fallopian tube, peritoneal cavity, and ovary [1,3,4]. It has been reported commonly in the gastrointestinal tract, especially in the stomach [3]. HCO is an extremely rare histologic subtype of ovarian cancer. Only a few cases of HCO have been reported in the literature [5]. This rare kind of tumor was described for the first time by Ishikura and Scully in 1987 [1]. Usually, it occurs in post-menopausal women with unilateral or bilateral ovarian masses and elevated serum AFP [4]. Microscopically, these tumors demonstrate cells arranged predominantly in sheets and contain a moderate to abundant amount of eosinophilic cytoplasm with pleomorphic nuclei [6]. They must be distinguished from metastatic HCC and other AFP-producing ovarian tumors including hepatoid yolk sac tumors (HYSTs), Sertoli–Leydig cell tumors, endometrial carcinoma, clear-cell carcinoma, lipid cell tumor, dysgerminomas, and undifferentiated carcinoma [1,7]. Commonly, it presents with signs and symptoms of an adnexal mass, such as progressive abdominal distension and lower abdominal pain. It can also be detected as an asymptomatic unilateral ovarian mass, rarer bilaterally [8]. Recurrence of HCO is common, and it often has a limited response to chemotherapy [5]. It occurs in an advanced stage (III-IV) [9] associated with a raised AFP [10]. Treatment for HCO and epithelial ovarian cancer is similar and includes cytoreductive surgery and further adjuvant therapy (e.i. carboplatin and paclitaxel-based therapy) [4]. For these reasons, it should be useful to establish diagnostic and prognostic markers for HCO to customize the treatment [5]. The first goal must be to improve scientific literature about the topic. Extremely rare, then, is the condition in which ovarian hepatoid carcinoma manifests as a metastasis of a primary hepatocarcinoma. In this latter case, recurrence after HCC usually appears within 2 years of initial treatment [11]. HCC, according to reports in the literature and with varying incidences, most frequently metastasizes to the peritoneal site, adrenal glands, lungs, and bone [12]. Instead, below, we will describe a very rare case of HCC recurrence in the ovary after 12 years from the first diagnosis.

## 2. Case Report

A 67-year-old woman was referred to our attention in February 2021 for evaluation of an adnexal mass. The patient, at the time, was followed by oncologists because she had a history of HCV-related hepatocellular carcinoma in 2009 (Edmonson e Stainer grade 2, pT2 N0 M0, G2, V1). She underwent surgical procedure for a left hepatectomy enlarged to the IV hepatic segment with cholecystectomy and subsequent Kehr drainage placement. At the time of the diagnosis of HCC in the liver site, her preoperatory AFP level was 8600 ng/mL, while the post-operative value was just 2.7 ng/mL. The serology for HBV and HCV viruses was negative. No other disease’s localizations were found and the genital tract was tumor free. Investigating the patient’s medical history, we knew she had familiarity with colic neoplasia and previous surgery for benign breast lumps. Her comorbidities were not linked with the disease: she was, in fact, affected by carotid atherosclerosis, diverticular colon disease, hyperthyroidism, and gastroesophageal disease. At the time of recurrence, she was completely asymptomatic: her liver function was unaffected, and blood works were normal. The follow-up was uneventful up until October 2020, when AFP was found raised (467 ng/mL), and CA125 became borderline (37.4 UI/mL). The oncologist indicated to perform a CT scan that showed the remaining liver had no lesions whatsoever, whilst her adnexa were both enlarged, especially on the left, where the ovary was 55 × 33 × 58 mm, with a solid cystic aspect with a few calcifications (on the right, the ovary was 37 × 31 mm) (Figure 1). A small fluid film was even found in the pelvis. No enlarged lymph nodes were located in the CT scan and the peritoneum was negative. Our next step was to perform a pelvic ultrasound examination. It reported that ovaries were indeed both enlarged: the left ovary was 62 × 23 × 34 mm and had a multilocular-solid cyst with a color score of 2 (IOTA adnex model risk malignancy 59.6%). It also described a modest pelvic effusion. Following a gyneco-oncological multidisciplinary meeting, the patient underwent surgery in March 2021. A laparoscopic left ovariectomy was performed with peritoneal biopsies at the level of the pelvic peritoneum and paracolic gutters. The ovary removed was sent for a frozen section. The extemporaneous histological examination found a tumor component with solid trabecular architecture composed of atypical cells with a high mitotic index which requires major definition at the definitive histological examination. In consideration of the patient’s history, we decided on a laparotomic approach with hysterectomy, oophorectomy, pelvic peritonectomy, and omentectomy. We obtained a tumor residue of 0 and there were no other macroscopic tumor localizations. The pathology reported on the left ovary a serous adenofibroma with adjacent a borderline serous tumor (TNM 2017: pT1a) and a hepatoid carcinoma with secondary repetitions in a nodule of the pelvic peritoneum (in Douglas and at the bladder’s peritoneal fold), and of the left paracolic gutter both. No other tumor extension was found. Immunohistochemistry investigations found positivity for CK 7 and focally for AFP. Common ovarian cancers’ immunoprofiles instead (ER, PR, p53, PAX8) were all negatives. The post-op recovery was uneventful, and the patient was discharged on the 3rd day after surgery. The case aroused undoubtedly great amazement because it required a careful multidisciplinary evaluation for the interpretation of the clinical data. On the one hand, it could have been interpreted as a primary hepatoid ovarian cancer, but this contrasted with the past history of HCC; on the other hand, the ovarian mass could represent a recurrence of the HCC, but this fought with the time elapsed from the first diagnosis (12 years). After also meticulously reviewing immunohistochemistry data, it was concluded for performing a PET scan and a hepatologic evaluation. The PET scan was negative for metastasis and the hepatologist found the ovarian hepatoid tumor most likely to be a recurrence, considering the relatively low level of AFP and the results of the molecular pathology examination. Our anatomo-pathologists played a primary role in the differential diagnosis: in fact, they performed a re-reading of the slides of the operative pieces compared with those of the surgery performed in 2009 for HCC, noting a consistent assonance of tumor cells. In addition, IHC confirmed the histologic diagnosis by giving a positive result for Hep PAR1, a marker strongly predictive of HCC, Arginase, and alpha-feto protein (Figure 2 and Figure 3). The patient continues with her 4 months follow up either with oncologists and gynecologists including clinical, laboratory, and instrumental data. The latest tumor markers assay of June 2022 showed: alpha fetal protein of 93.7 IU/mL (increasing) and CA125 of 18.3 IU/mL. Control upper abdomen MRI shows no images suggestive of recurrence of disease. The patient is currently asymptomatic.

## 3. Discussion

Primary HCO is a rare, aggressive tumor that shares clinical and pathologic features with HCC and is thought to have 2 histogenic origins—i.e., superficial epithelium and germ cells, although neither of these is supported by conclusive evidence [13]. Hepatocarcinoma usually spreads by the hematogenous or lymphatic route, or by direct invasion of adjacent organs. Common sites of metastasis are lungs, peritoneum, bones, and adrenal glands [14]. The ovary is a rare site of metastasis for hepatocellular carcinoma, with less than 20 cases reported in the literature up to now [12,15]. Recurrence generally occurs within 2 years of the first diagnosis. The prognosis is still poor, with median survival after diagnosis of extrahepatic metastasis of 8 months [16]. In 50% of the cases, peritoneum localizations were reported, mostly in patients with a bilateral ovarian tumor [15]. HCC rarely metastasizes to the ovary. Moreover, metastatic HCC should be distinguished from a hepatoid yolk sac tumor and a primary hepatoid carcinoma of the ovary. Hepatoid carcinomas of the ovary and hepatoid yolk sac tumors are suspected when there is no clinical or operative evidence of HCC. In the case presented herein, the patient had a history of HCC and there was no evidence of hepatoid carcinomas in other abdominal organs, as suggested by Lee et al. [12]. Renzulli et al., in their 2022 literature review, describe a very interesting diagnostic chance for HCC: they identify radiologic features that can to date provide the clinician with a diagnosis of certainty [17]. Specifically, as pathognomonic of this liver tumor, they identify dynamic image studies (such as computed tomography or magnetic resonance imaging) and the typical dynamic image of HCC showing intense arterial uptake followed by “washout” of contrast in the venous and/or delayed phases. Their careful examination, however, was of little help in our case since the treatment was mainly useful on cirrhotic livers having nodular or other lesions. Our patient had neither a history of cirrhosis nor evidence of recurrent disease in the residual liver. Histology plays a key role in differential diagnosis: for example, in the literature, it is reported that bile pigment has been described in hepatoid carcinoma of the stomach and renal pelvis [18]. At the same time, Hep Par 1, a highly specific marker of HCC (found positive in our patient), can help in the differential diagnosis between HAC and HCC, but unfortunately, some HAC tumors may still show positive staining. This certainly makes us consider that more studies are needed to standardize diagnosis and next, treatment itself. CK7 was found positive in a very low percentage of the few reported cases of HAC, considered out of a total to be gastric in origin [2]. In our patient, it was positive, leaning more toward a case of HCC. Regarding the emerging but still experimental role of PLUNC staining for differential diagnosis between HAC and HCC, in 52 cases of HCC, it was never found; and for the HAC-positive cases, they were all primary gastric cancers. It follows that in addition to having to validate its usefulness, its universality of applicability will have to be defined regardless of the primary tumor [2]. In their case series, Nguyen et al. concluded instead that HAC is an extremely rare neoplasm that remains a differential diagnosis in AFP-producing liver lesions without the imaging features typical of HCC [2,18]. Additionally, because of this assumption, our case differed in the absence of underlying cirrhotic lesions at the time of relapse. They also report that the form of primary adnexal HAC, in the underlying rarity of the disease (with an incidence of 0.014 per 100,000 people in the Asian population), is reported in only 4% of cases, compared with 33% for colorectal, 21% for neuroendocrine tumors of various origins, and 17% pancreatic. This is a finding further in favor of diagnosing ovarian recurrence of hepatocarcinoma rather than hepatoid primitiveness. Still, the serum level of AFP was > 200 ng/mL—precisely, in our patient, it was 467 ng/mL—which, according to international guidelines, is a standard criterion for the diagnosis of HCC [19]. All these findings, combined with the patient’s strongly indicative history of previous HCC, greatly influenced the clinical judgment in favor of a late ovarian recurrence of HCC. In Table 1, the main features that guided the multidisciplinary differential diagnosis between HAC and HCC are summarized.

In our case, the story of a previous hepatocarcinoma suggested the lesions to be a recurrence. In order to resolve the numerous doubts that the evidence of medicine requires us to clarify, in addition to immunohistochemistry, an anatomical-pathological study comparing the current surgical pieces with the hepatocarcinoma removed in 2009 was necessary. It thus emerged that the two lesions were morphologically overlapping. Furthermore, molecular analyses were conducted to compare the hepatic HCC and the ovarian lesion with the same molecular fingerprint. Moreover, the liver lesion of first diagnosis (2009) was described as high-risk from the outset as it had biliary infiltration and vascular neoplastic thrombosis. Indeed, considering the rarity of such a long time to relapse, we needed more data. Molecular data could surely help the diagnosis. Among immunoprofiles, the f.e. antihepatocyte antibody is not useful to differentiate between ovarian tumors with hepatoid phenotypes, such as POHC and metastatic HCC. The CK 7+/20− profile may suggest that this favors an ovarian tumor arising from a common surface epithelium over non-mullerian tumors. Although most HCC is negative for CK 7, some works reported a rate of CK7 positivity of almost 30% for HCC and some even associated this positivity with a worse prognosis [20,21,22,23]. On immunohistochemical analysis, positivity for the antihepatocyte antibody, hepatocyte-specific CK (8 and 18), and biliary-specific CK (7 and 19) panels may help to support a diagnosis of primary ovarian hepatoid carcinomas [20]. Our patient had positivity for CK 7 and AFP (Figure 1 and Figure 2). However, the differentiation between hepatic HCC and the ovarian hepatoid tumor is not straightforward, as morphologically and immunohistochemically the lesions are very similar. Therefore, what discriminates is the clinic and the overall multidisciplinary evaluation of the case. In the literature, there is no scientifically significant evidence of recurrence of HCC in the ovarian site and isolated cases are still reported. Notwithstanding this, in their 2011 paper, Lee et al. include a brief review of the literature, in which they report approximately 10 cases of HCC recurrence in the ovarian site with the respective time of occurrence [12]. It emerges that the maximum time was 2–8 years from diagnosis and surgical treatment [12]. In their review of 11 cases of metastatic HCC to the ovary, all but one case showed elevated serum levels of AFP. Among the reported cases of metastatic HCC to the ovary, serum AFP levels decreased after surgery and increased again at recurrence [12]. Therefore, serum AFP may be a useful indicator of recurrent HCC in the ovary and should be closely observed. Indeed, the authors conclude that, despite its rarity, AFP elevation in the presence of a pelvic mass, of no other plausible origin, should raise suspicion for the site of ovarian metastasis from HCC [12]. Li Liu et al., in their 2012 paper, go on to discuss the microscopic and immunohistochemical features of HCO [12]. They emphasize that a diffusely positive staining for CK7 is considered to be highly consistent with HCO, in contrast to ovarian recurrence of HCC. A differential diagnosis between the two histotypes is certainly a challenge: to date, there are no clear tools to distinguish the two tumors [12]. In the present case, the patient developed ovarian lesions after 12 years from the primary treatment, which is the longest time elapsed from first diagnosis to recurrence. The longest time reported in the literature for ovarian HCC recurrence is two years after surgical treatment at first diagnosis with orthotopic liver transplantation [11]. We report this case because it was very difficult to correlate clinical history with histological diagnosis, given the lack of scientific literature. So, a multidisciplinary meeting was essential for the gynecologists to establish the correct diagnostic classification and to decide the subsequent therapeutic iter. In the current state of evidence, a possible differentiation between histotypes therefore lies in assembling clinical, oncological, histological, and IHC data. All these elements together and related to the literature were brought into play to manage this interesting case.

## 4. Conclusions

Hepatoid carcinomas of the ovary must be distinguished from ovarian metastasis of hepatocarcinoma. We need more scientific data to follow evidence-based medicine in this field. In this regard, it could be useful to start with implementing molecular biology and IHC to help clinicians and pathologists to distinguish these two rare conditions. Knowledge about the pathologies, however, must be improved: actually, because of their rarity, there are only few case reports or case series. For this reason, for the moment, we can assert our case is the first signaled to be an HCC relapsed after 12 years. Probably, though, we only need to learn more about this intriguing topic.

## Figures and Tables

**Figure 1 jcm-12-02468-f001:**
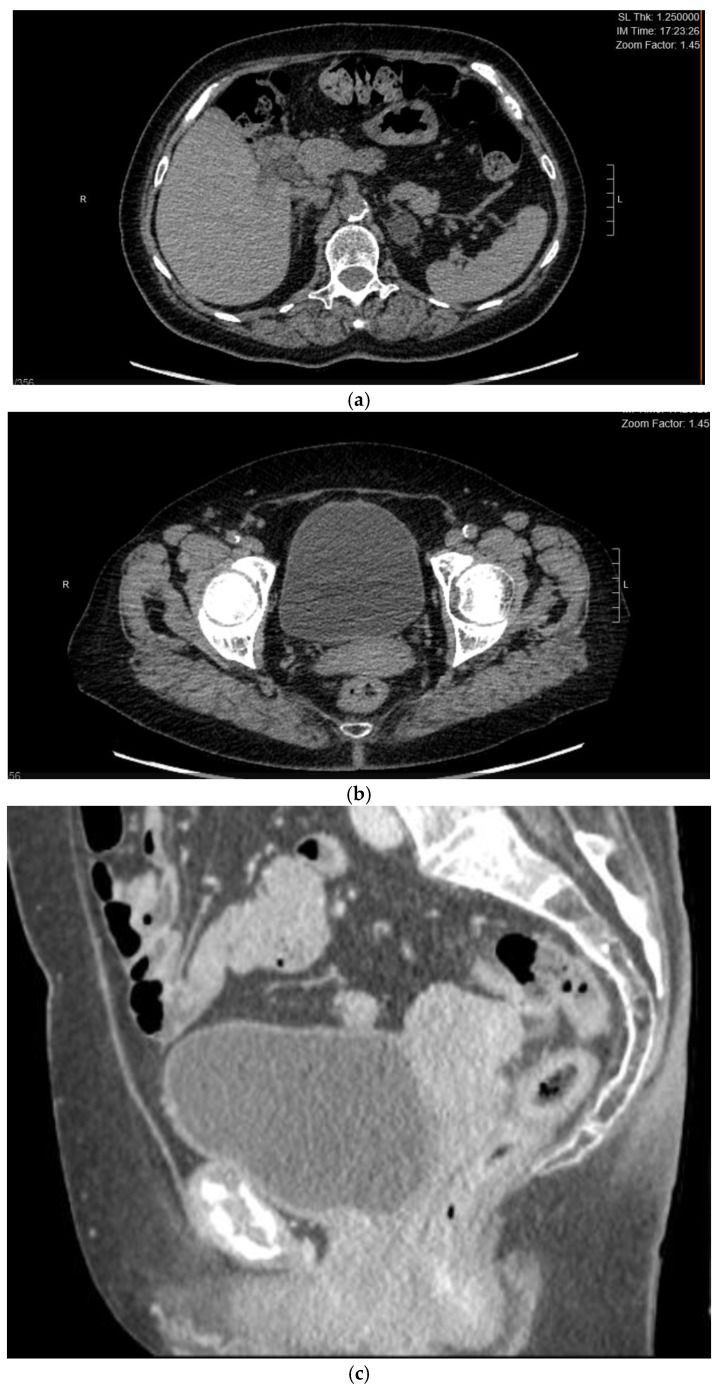
(**a**) Preoperative CT cross-section at the liver level, free of pathological lesions. (**b**,**c**) Multilocular solid left adnexal mass, recurrence of HCC, seen in sagittal and transverse scans.

**Figure 2 jcm-12-02468-f002:**
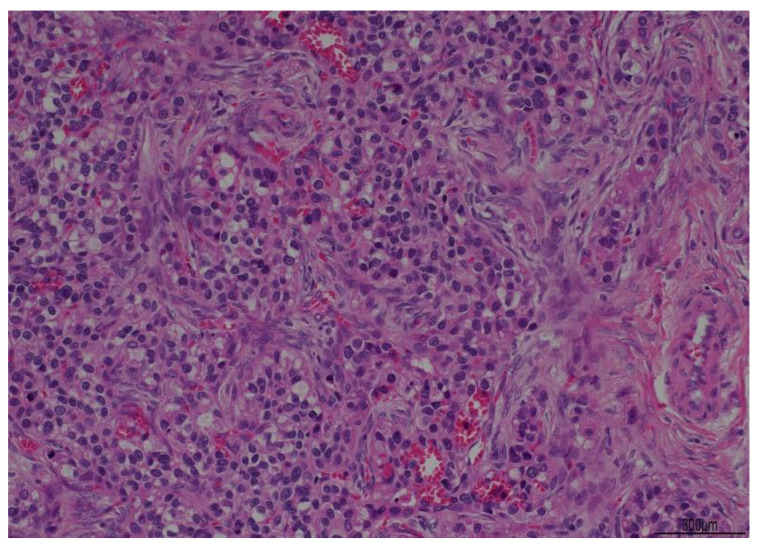
Histological examination slides in hematoxylin-eosin at 10× magnification.

**Figure 3 jcm-12-02468-f003:**
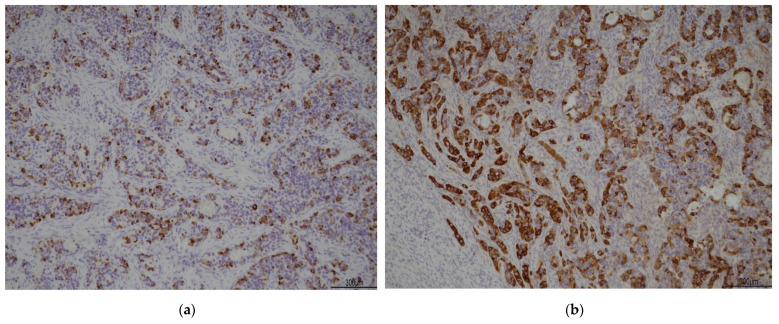
(**a**) hepatocyte 10×; (**b**) alfa-feto protein 10×.

**Table 1 jcm-12-02468-t001:** Main features of the differential diagnosis between HAC and HCC.

Site of primary tumor	Ovary (only 1 case of HAC cited in Sheng Su et al. review)
Remote pathological history	HCC in 2009 treated surgically
Tumor markers	AFP > 200 ng/mL
IHC	- Cytokeratin 7: positive
	- PAX8: negative
	- Hepatocyte: focally positive cells
	- Glypican 3: positive
	- Arginase: focally positive
	- WT1: negative

## Data Availability

This paper was presented as poster at the SIGO 2021 Congress that was held in Sorrento (NA) on December 2021 but never published.

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
