# Peer review of "A Rare Case of Hepatocellular Carcinoma Recurrence in Ovarian Site after 12 Years Mimicking a Hepatoid Adenocarcinoma: Case Report"

_jcm, 2023, doi:10.3390/jcm12072468_

Round 1

Reviewer 1 Report

Thank you for giving me the opportunity to review this manuscript. The case report describes a 67-year-old woman with history of hepatocellular carcinoma (Edmonson e Stainer grade 2). She presented with bilateral adnexal tumors, pelvic ascites, and elevated AFP levels. The patient underwent hysterectomy, bilateral oophorectomy, peritonectomy, and omentectomy. It has been difficult to differentiate between a recurrent hepatocellular carcinoma and a primary hepatoid carcinoma of the ovary on the basis of histological and immunological findings since both tumors share a number of similar features.

This manuscript is well written. An important piece of information has been provided about a rare tumor in this study. However, I would recommend that the author add more detail about postoperative treatment. What type of chemotherapy or adjuvant treatment did the patient receive?

Author Response

Dear reviewer, we thank you for your comments and suggestions.
In response, first of all, I would like to emphasize that the histologic slides were carefully and adequately reviewed to confirm the diagnosis.
Subsequently, the patient's clinical case was thoroughly discussed in both gyneco-oncologic and gastro-enterologic multidisciplinary meetings. She was therefore re- evaluated by the hepatology colleagues who had treated her in 2009 for the first diagnosis of HCC who in turn leaned toward an interpretation in favor of HCC recurrence rather than primary ovarian hepatoid tumour. After appropriate multidisciplinary review of the instrumental examinations and intraoperative findings, and in particular the tumor residual reached of 0 and negative PET/CT imaging for further localization, it was agreed for close clinical, instrumental, and laboratory follow-up with alpha-feto protein assay, a marker that was indicative of disease recurrence, without posing indications for further adjuvant therapies.
Attached you will find the edited version of the manuscript.

Reviewer 2 Report

The case report entitled "A rare case of hepatocellular carcinoma recurrence in ovarian 2
site after 12 years: case report" by Restaino et al, presents a rare case of isolated recurrence of HCC in the ovary after a long time gap of 12 years.

The authors mention that the uniqueness of the case lies in " rarity of the pathology, the lack of scientific evidence and the difficulty in differentiating ovarian primitivity from HCC metastasis".

I have a few comments:

1) the treatment offered was " She therefore underwent extemporaneous examination- 30
guided radical surgery". Kindly explain.

2) IHC was CK7+, AFP focal. What was the report status of CK20, HepPAR1, Arginase. These markers are routinely employed to diagnose HCC on biospy.

3.Left para colic gutter and POD had deposits of HCC. In that case, after this radical surgery, the tumor board should have considered systemic therapy. Atezolizumab/Bevacizumab or Lenvatinib. This needs to be mentioned.

4. Expand IOTA adnex model

5. What markers were done to rule out a germ cell tumor of ovary, although this is highly unlikely in a 67 year lady?

6. The value of above IHC should be added to the discussion. The role of serum tumor marker PIVKA also needs to be highlighted in a case of HCC.

7. If we see closely, the case is ovarian + peritoneal metastases, not isolated ovary.

8. The manuscript needs some English language editing.

Author Response

Dear reviewer, we thank you for your comments and suggestions. Below we try to give you more clarifications.

About the first comment: 1. the treatment offered was " She therefore underwent extemporaneous examination- 30 guided radical surgery". Kindly explain.

The surgery, performed as a diagnostic LPS with subsequent LPT conversion, was conducted with initial tumour mass site adnexiectomy with subsequent intraoperative sending of extemporaneous histologic examination. Histologic diagnosis guided subsequent surgery with intent and achievement of residual tumor of 0.

  1. IHC was CK7+, AFP focal. What was the report status of CK20, HepPAR1, Arginase. These markers are routinely employed to diagnose HCC on biospy.

Regarding IHC: thanks for the question. CK20 was negative; Hep PAR1 and Arginase positive.

  1. Left para colic gutter and POD had deposits of HCC. In that case, after this radical surgery, the tumor board should have considered systemic therapy. Atezolizumab/Bevacizumab or Lenvatinib. This needs to be mentioned.

The patient after radical surgery and appropriate histologic review of the slides, was evaluated in multidisciplinary both gyneco and gastroenteric oncology boards who did not place an indication for adjuvant therapy; postoperative PET scan was negative and the hepatology colleagues who had had her in 2009, agreed for close follow-up. We have taken care to include a clarification about this in the text.

  1. Expand IOTA adnex model.

“left ovary was 62 x 23 x 34 mm and had a multilocular-solid cyst with a color score of 2 (IOTA adnex model risk malignancy 59.6%)”.

The IOTA Adnex Model is a standardized ultrasound diagnostic score in the ultrasound diagnosis of adnexal masses. It consists of a series of items, such as the patient's age at the time of the examination, performance in a referral center in Gynecologic Oncology; maximum diameter of the formation and major solid part expressed in mm, possible presence of > 10 loculi, number of papillae, presence of shadow cones, ascites, and CA125 blood assay.
In the patient's case: this was the IOTA description: “Solid multilocular formation with a cystic component with inhomogeneous content, vascularized CS2 measuring 62 x 23 x 34 mm”

  1. What markers were done to rule out a germ cell tumor of ovary, although this is highly unlikely in a 67 year lady?

The germ cell tumor (yolksac in this case), as you pointed out, is more typical of young age (< 30 years) and often has multiple morphological features in the same lesion, with characteristic Schiller- Duval bodies that were not there in our case (although hepatoid yolksac may be pure). Its markers, which we did, are alpha fetoprotein, glipican and focally HepPar1, which are analogous to those of hepatocellular carcinoma (the latter generally also expresses arginase, which in our case was in fact positive). The differential diagnosis with hepatocellular carcinoma metastasis is generally made more by clinical history than by histological and immunohistochemical features in this case.

6. The value of above IHC should be added to the discussion. The role of serum tumor marker PIVKA also needs to be highlighted in a case of HCC.

PIVKA assay is not something we do in our anatomic pathology but I think it is dosed in laboratory analysis and I don't know if it is done here, we in IHC, we evaluated alpha fetoprotein. We have provided more information about IHC in the discussion.

  1. If we see closely, the case is ovarian + peritoneal metastases, not isolated ovary.

We thank you for the clarification. You are right, at the final histologic diagnosis, peritoneal biopsies were also positive for localizations of disease. Our intent, however, in accordance with literature data that tell us that peritoneal dissemination of hepatocellular carcinoma (HCC) is yes, rare, with an incidence of 2-6%, according to others between 5-15%, but certainly better known and studied than ovarian dissemination, which to date, figures in few and only case reports.

Spiliotis J, Nikolaou G, Kopanakis N, et al., Hepatocellular Carcinoma Peritoneal Metastasis: Role of Cytoreductive Surgery and Hyperthermic Intraperitoneal Chemotherapy (HIPEC). Gulf J Oncolog. 2017 May;1(24):20-23.

Da Fonseca LG, Leonardi PC, Hashizume PH, et al., A multidisciplinary approach to peritoneal metastasis from hepatocellular carcinoma: clinical features, management and outcomes. Clin Exp Hepatol. 2022 Mar;8(1):42-48.

Kwak MS, Lee JH, Yoon JH, et al., Risk factors, clinical features, and prognosis of the hepatocellular carcinoma with peritoneal metastasis. Dig Dis Sci. 2012 Mar;57(3):813-9.

  1. The manuscript needs some English language editing.

We improved the style and English.

Attached you will find the edited version of the manuscript with further details.

Reviewer 3 Report

Dear Authors

I would like to thank you for the opportunity of reviewing this interesting paper that is focused on a very challenging topic also in the daily clinical practice. The present paper reports a rare case of hepatocarcinoma’s recurrence in the ovary after 12 years from the first diagnosis mimicking a hepatoid adenocarcinoma. I think that this paper can result endearing to many readers since provide a interesting example on how discriminate an extrahepatic HCC from a hepatoid adenocarcinoma. This paper is pleasurable to read, although it suffers from some limitations that Authors can easily adjust in order to slightly improve their article making it more eligible for this important Journal. 

First of all, although language used is appropriate, I (I am not a native English speaker) recommend to Authors to obtain a certified native speaker with proficiencies in the scientific-medical field to properly complete this paper, because several sentences are not completely fluent. Moreover, I recommend making a further revision of the text to fix some typing errors.

Personally, from a stylistic point of view, I think the title could be improved, for example: “A rare case of hepatocellular carcinoma recurrence in ovarian site after 12 years mimicking a hepatoid adenocarcinoma: case report”.

Authors did not correctly reported keywords from MeSH Browser. In particular, I checked for example “ovarian metastasis of hepatocarcinoma” on MeSH Browser and this is not a KW. This is important, in my personal opinion, in order to increase the traceability of this paper (and consequently the possibility of the Journal to be cited by Readers and Stakeholders). I suggest the check of all KW and use only those that are present in MeSH Browser.

The abstract needs revision, adding more information regarding the previous HCC (stage, therapy, etc.). Moreover, please use the abbreviations after the first mention (do not re-write “hepatocellular carcinoma” but use “HCC). Finally, please check this sentence: “This was a time longer than every other case reported in the literature to date”, which sounds a bit odd.

Although the introduction appears concise and interesting to read, I believe that the authors could improve this section focusing more on the results and improving the scientific soundness. For example, “Hepatoid adenocarcinoma is a tumour that resembles both histologically and cytologically a hepatocarcinoma (HCC) in a patient with a healthy liver not involved by the disease”. Please correct “healthy” with “non-cirrothic” (even in the abstract and the entire text) and specify that hepatoid adenocarcinoma is, by definition, an extrahepatic adenocarcinoma with hepatocyte differentiation. [World J Gastroenterol. 2013 Jan 21;19(3):321-7. doi: 10.3748/wjg.v19.i3.321]

Lines 65-67: this sentence is odd, please check.

In the “Case Report” section, “The oncologist indicated to perform a CT scan that showed the remaining liver had no lesions whatsoever whilst her adnexa were both enlarged, especially on the left where the ovary was 55x33x58 mm”. Please provide CT scans of both liver and ovary and a brief description of the imaging features of the ovarian lesion. If available, please add also the US images.

Lines 115-118, “The PET scan was negative for metastasis and the hepatologist found the ovarian hepatoid tumor most likely to be a recurrence, considering the relatively low level of AFP and the results of the molecular pathology examination.” This section needs to be expanded and the motivations behind the higher probability of diagnosis of HCC must be explained more extensively, describing what molecular markers support the diagnosis of HCC rather than hepatoid adenocarcinoma. 

In the “Discussion” section, please, be very strict in what you describe, and try to avoid repetitions. Moreover, please be more clearer regarding the characteristics that allow to make the differential diagnosis between the two entities. For example, diagnosis of HCC should be based on dynamic image studies (such as computed tomography or magnetic resonance imaging) and the typical dynamic image of HCC showing intense arterial uptake followed by “washout” of contrast in the venous and/or delayed phases [Histol Histopathol. 2022 Dec;37(12):1151-1165. doi: 10.14670/HH-18-487]. Moreover, in the differential diagnosis between HAC and HCC for AFP-producing liver tumors, HAC is strongly suggested if AE1/AE3, CK18 and C19 stains show strong positive findings. Similarly, Hep Par 1, a highly specific marker of HCC, may help in the differential diagnosis between HAC and HCC, but some HAC tumors can still present with positive staining. Finally, positive PLUNC staining was proposed as a novel marker that might well distinguish HAC from HCC. [GH Open. 2022 Sep 8;6(10):727-729. doi: 10.1002/jgh3.12813] [World J Gastroenterol. 2013 Jan 21;19(3):321-7. doi: 10.3748/wjg.v19.i3.321]. Please, add all the aforementioned references in the text and provide a brief table with the main clinical, imaging and laboratory findings for differential diagnosis between HCC and hepatoid adenocarcinoma.

Last but not least, please follow the Journal guidelines for reference in the text (reference numbers should be placed in square brackets [] and placed before the punctuation) and in the Reference section (Author 1, A.B.; Author 2, C.D. Title of the article. Abbreviated Journal Name YearVolume, page range)

Best regards, 

Author Response

Dear reviewer, we thank you for your precious comments and suggestions. Here, we summarize the answers to your questions.

  • Personally, from a stylistic point of view, I think the title could be improved, for example: “A rare case of hepatocellular carcinoma recurrence in ovarian site after 12 yearsmimicking a hepatoid adenocarcinoma: case report”.

Thank you for the suggestion: indeed it might make the title more appealing for a reading. We have edited it!

  • Authors did not correctly reported keywords from Mesh Browser.

These were the previous keywords: hepatoid carcinoma1, ovarian metastasis of hepatocarcinoma2, histological differential diagnosis3, multidisciplinary approach4, case report5

We changed them consulting the Mesh Browser site: thank you for the note. These are the new keywords: Liver neoplasms1, adnexal diseases2, genital neoplasms female3, diagnosis differential4, case reports5.

  • The abstract needs revision, adding more information regarding the previous HCC (stage, therapy, etc.). Moreover, please use the abbreviations after the first mention (do not re-write “hepatocellular carcinoma” but use “HCC). Finally, please check this sentence: “This was a time longer than every other case reported in the literature to date”, which sounds a bit odd.

This is our new abstract.

Hepatoid carcinoma of the ovary (HCO) is a tumor that resembles both histologically and cytologically hepatocarcinoma (HCC) in a patient with a non cirrothic liver not involved by the disease (1). Hepatoid carcinoma is an extremely rare histologic subtype of ovarian cancer and should be distinguished from metastatic HCC. Here we report the rare case of a 67-year-old woman with ovarian recurrence of HCC 12 years after first diagnosis. The patient was being followed by oncologists because she had been diagnosed with HCV-related HCC (Edmonson and Stainer grade 2, pT2 N0 M0, G2, V1) in 2009. She had undergone surgery for enlarged left hepatectomy to the 4th hepatic segment with cholecystectomy and subsequent placement of a Kehr drain. The preoperative alpha-fetoprotein (AFP) level was 8600 ng/ml, while the postoperative value was only 2.7 ng/ml. At the first diagnosis, no other localization of the disease, including the genital tract, was found. At the time of recurrence, however, the patient was completely asymptomatic: her liver function was within normal limits with negative blood indices, except for an increased blood dosage of AFP (467 ng/ml), and CA125 had become borderline (37.4 IU/ml). The oncologist placed an indication for a thoracic abdominal CT scan, which showed that the residual liver was free of disease, but the presence of a formation with a solid-cystic appearance and with some calcifications at the left adnexal site. The picture was confirmed on level II gynecologic ultrasonography.
The patient then underwent a radical surgery of hysterectomy, oophorectomy, pelvic peritonectomy, and omentectomy by a laparotomic approach, with sending of intraoperative extemporaneous histological examination, obtaining a RT=0. Currently, the patient continues her gyneco-oncologic follow-up simultaneously clinically, laboratory and instrumentally every 4 months. Our study currently represents the longest elapsed time interval between first diagnosis and disease recurrence, as evidenced by current data in the literature. This was a rather unique and difficult clinical case because of the rarity of the disease, the lack of scientific evidence, and the difficulty in differentiating the primary hepatoid phenotype of the ovary from an ovarian metastasis of HCC. Serious multidisciplinary meetings for proper interpretation of clinical and anamnestic data, with the aid of immunohistochemistry (IHC) on histological slides were essential for case management.

We paid more attention to non-repetition of sentences and to insert abbreviations appropriately.

  • Although the introduction appears concise and interesting to read, I believe that the authors could improve this section focusing more on the results and improving the scientific soundness. For example, “Hepatoid adenocarcinoma is a tumour that resembles both histologically and cytologically a hepatocarcinoma (HCC) in a patient with a healthy liver not involved by the disease”. Please correct “healthy” with “non-cirrothic” (even in the abstract and the entire text) and specify that hepatoid adenocarcinoma is, by definition, an extrahepaticadenocarcinoma with hepatocyte differentiation.[World J Gastroenterol. 2013 Jan 21;19(3):321-7. doi: 10.3748/wjg.v19.i3.321]

Thank you for your comment. You will find more details on the value of the results in light of the scientific evidence with suggested bibliographic references included beyond that as well.

  • Lines 65-67: this sentence is odd, please check.

You will find the reported sentence modified as follows: “HCC, according to reports in the literature and with varying incidences, most frequently metastasizes to the peritoneal site, adrenal glands, lungs, and bone (11). Instead, below we will describe a very rare case of HCC recurrence in the ovary after 12 years from the first diagnosis”.

  • In the “Case Report” section, “The oncologist indicated to perform a CT scan that showed the remaining liver had no lesions whatsoever whilst her adnexa were both enlarged, especially on the left where the ovary was 55x33x58 mm”. Please provide CT scans of both liver and ovary and a brief description of the imaging features of the ovarian lesion. If available, please add also the US images.

In the revised paper, attached below, you'll find the required CT scans. Regarding ultrasound imaging, the patient's medical record has already been filed. Therefore if needed, we will request access but the procedure will take some time.

These are the descriptions of the CT scans included in the manuscript. Figure1: Preoperative CT cross-section at the liver level, free of pathological lesions.

Figures 2 and 3: Multilocular solid left adnexal mass, recurrence of HCC, seen in sagittal and transverse scans.

  • Lines 115-118, “The PET scan was negative for metastasis and the hepatologist found the ovarian hepatoid tumor most likely to be a recurrence, considering the relatively low level of AFP and the results of the molecular pathology examination.” This section needs to be expanded and the motivations behind the higher probability of diagnosis of HCC must be explained more extensively, describing what molecular markers support the diagnosis of HCC rather than hepatoid adenocarcinoma. 

Our anatomo-pathologists played a primary role in the differential diagnosis: in fact, they performed a re-reading of the slides of the operative pieces compared with those of the surgery performed in 2009 for HCC, noting a consistent assonance of tumor cells. In addition, IHC confirmed the histologic diagnosis by giving a positive result for Hep PAR1, a marker strongly predictive of HCC, arginase and alpha-feto protein.
We refer to the paper, for more details.

  • In the “Discussion” section, please, be very strict in what you describe, and try to avoid repetitions. Moreover, please be more clearer regarding the characteristics that allow to make the differential diagnosis between the two entities. For example, diagnosis of HCC should be based on dynamic image studies (such as computed tomography or magnetic resonance imaging) and the typical dynamic image of HCC showing intense arterial uptake followed by “washout” of contrast in the venous and/or delayed phases [Histol Histopathol. 2022 Dec;37(12):1151-1165. doi: 10.14670/HH-18-487]. Moreover, in the differential diagnosis between HAC and HCC for AFP-producing liver tumors, HAC is strongly suggested if AE1/AE3, CK18 and C19 stains show strong positive findings. Similarly, Hep Par 1, a highly specific marker of HCC, may help in the differential diagnosis between HAC and HCC, but some HAC tumors can still present with positive staining. Finally, positive PLUNC staining was proposed as a novel marker that might well distinguish HAC from HCC. [GH Open. 2022 Sep 8;6(10):727-729. doi: 10.1002/jgh3.12813] [World J Gastroenterol. 2013 Jan 21;19(3):321-7. doi: 10.3748/wjg.v19.i3.321]. Please, add all the aforementioned references in the text and provide a brief table with the main clinical, imaging and laboratory findings for differential diagnosis between HCC and hepatoid adenocarcinoma.

Renzulli et al., in their 2022 literature review, describe a very interesting diagnostic chance for HCC: they identify radiologic features that can to date provide the clinician for a diagnosis of certainty. Specifically as pathognomonic of this liver tumor, they identify dynamic image studies (such as computed tomography or magnetic resonance imaging) and the typical dynamic image of HCC showing intense arterial uptake followed by "washout" of contrast in the venous and/or delayed phases. Their careful examination, however, was of little help in our case since the treatment was mainly on cirrhotic livers having nodular or other lesions. Our patient had neither a history of cirrhosis nor evidence of disease in the residual liver.
Regarding the IHC mentioned, thanks for the clarification. In our experience, CK AE1/AE3 is rather irrelevant since we are dealing with a carcinoma, also confirmed by positivity for CK7 and CK 19 is generally ubiquitous so it doesn't add much in differential diagnosis; however if they are essential we can do them now.

In their case series, Nguyen et al. concluded that HAC is an extremely rare neoplasm that remains a differential diagnosis in AFP-producing liver lesions without the imaging features typical of HCC. Also because of this assumption, our case differed in the absence of underlying cirrhotic lesions. They also report, that the form of primary adnexal HAC, in the underlying rarity of the disease (with an incidence of 0.014 per 100000 people in the Asian population), is reported in only 4% of cases, compared with 33% for colorectal, 21% for neuroendocrine tumors of various origins, and 17% pancreatic. This is a finding further in favor of diagnosing ovarian recurrence of hepatocarcinoma rather than hepatoid primitiveness.
Still, the serum level of AFP was > 200 ng/ml - precisely in our patient it was 467 ng/ml - which, according to international guidelines, is a standard criterion for the diagnosis of HCC [bruix].
CK7 was found positive in a very low percentage of the few reported cases of HAC, considered out of a total to be gastric in origin [sheng su]. In our patient, it was positive, leaning more toward a case of HCC. All these findings, combined with the patient's strongly indicative history of previous HCC, greatly influenced the clinical judgment in favor of a late ovarian recurrence of HCC.
Emerging but still experimental role of PLUNC staining for differential diagnosis between HAC and HCC. In 52 cases of HCC, it was never found; and for the HAC-positive cases, they were all primary gastric cancers. It follows that in addition to having to validate its usefulness, its universality of applicability will have to be defined regardless of the primary tumor [sheng on]. However, in our center, we do not have immunohistochemistry for PLUNC.

 All bibliographic references suggested have been included.

Site of primary tumour

Ovary (only 1 case of HAC cited in Sheng Su et al. review)

Remote pathological history

HCC in 2009 treated surgically

Tumor markers

AFP > 200 ng/ml

IHC

- Cytokeratin 7: positive

- PAX8: negative

- Hepatocyte: focally positive cells

- Glypican 3: positive

- Arginase: focally positive

- WT1: negative

  • Last but not least, please follow the Journal guidelines for reference in the text (reference numbers should be placed in square brackets [] and placed before the punctuation) and in the Reference section (Author 1, A.B.; Author 2, C.D. Title of the article. Abbreviated Journal NameYearVolume, page range)

I thank you for the suggestion. We have arranged the bibliography with the additions rightly suggested.

Attached you will find the edited version of the manuscript with further details.

Round 2

Reviewer 3 Report

Authors addressed raised points appropriately.